# Use of Visual Dashboards to Enhance Pharmacy Teaching

**DOI:** 10.3390/pharmacy9020093

**Published:** 2021-04-23

**Authors:** Andrew Bartlett, Carl R. Schneider, Jonathan Penm, Ardalan Mirzaei

**Affiliations:** 1School of Pharmacy, Faculty of Medicine and Health, The University of Sydney, Camperdown, NSW 2006, Australia; carl.schneider@sydney.edu.au (C.R.S.); jonathan.penm@sydney.edu.au (J.P.); ardalan.mirzaei@sydney.edu.au (A.M.); 2Department of Pharmacy, Prince of Wales Hospital, Randwick, NSW 2031, Australia

**Keywords:** visual analytics, Dashboards, TPACK, Open leaner model, action-based research

## Abstract

Teaching large cohorts of pharmacy students with a team of multiple tutors in a feedback intensive course poses challenges in relation the amount of data generated, data integrity, interpretation of the data and importantly application of the insights gained from the data. The dispensing and counselling course in the third year BPharm at the University of Sydney has implemented the USyd Pharmacy Dashboard, developed to address these challenges following the Technological Pedagogical Content Knowledge Framework (TPACK) to integrate technology into teaching. The dashboard was designed to improve the student experience through more consistent feedback, gain insights to improve teaching delivery and provide efficiencies in maintaining data integrity. The tool has been developed using an action-based research approach whereby ideas are put into practice as the means to further develop the idea and improve practice. Refinement of the USyd Pharmacy Dashboard over three years has shown improvements in teaching delivery as teachers can respond to emerging trends. Student performance and satisfaction scores have increased, mainly due to improved consistency between tutors and improved delivery of feedback. Time involved with administrative tasks such as data maintenance is reduced. Opportunities for further refinements such as real time benchmarking and developing an open learner model have become apparent.

## 1. Background and Purpose

University teaching is increasingly becoming more dependent on technology on a day-to-day basis [1]. Ranging from commercial learning management systems such as Canvas^©^ and Blackboard^®^ to self-developed technologies such as the Student Relationship Engagement System (SRES) developed at The University of Sydney [2]. These tools collect volumes of data. However, it is imperative to know what to do with that data, how to access relevant information and adapt teaching strategies to improve student outcomes.

Teaching pharmacy dispensing and patient counselling to a large cohort of students over multiple days with multiple tutors presents many challenges. The questions raised have been not only of quality assurance, ensuring a consistent approach to the tasks and as well as consistency of the feedback provided, but also how to communicate that approach to the team delivering the course and provide student’s feedback [3].

Teaching the skill of patient counselling can involve grey areas. The tutors’ personal preferences or personality, as well as practice history, can influence what they see as important for students to learn. A consistent message needs to be conveyed to students through these teachers under the direction of the course coordinator to avoid confusing students about expected standards.

Understanding how technology, content knowledge and pedagogy interact is critical in the modern tertiary educational environment. This relationship may be explained by the Technological Pedagogical Content Knowledge Framework (TPACK) as a tool to aid the integration of technology into teaching [4]. The authors in a 2014 article highlight that each of these do not exist in isolation but must be developed together [4]. This is an important consideration, and highlights the need for education of teachers, tutors, or demonstrator in the use of the technologies, learning styles, feedback provision and the topic area.

### Dashboards

The tertiary education environment generates an enormous amount of student-generated data, ranging from attendance to marks, comments, and content. This results in a complex array of data structures, requiring significant time and expertise to interpret. Visual analytics is the generation of interactive visualizations from data to allow human intuition to interpret and gain insight from perceived patterns in the data [5]. The identification of patterns forms part of the decision-making process for evaluating performance or identifying gaps [6,7]. A sense-making loop is then created, whereby the user interacts and manipulates the visual data gaining further insights into both the visual representation and the data itself [7]. A visualization tool that can be used as part of decision making is a dashboard [8].

Dashboards in education are considered part of learning analytics, a discipline defined as ‘the measurement, collection, analysis, and reporting of data about learners and their contexts, for the purposes of understanding and optimizing learning and the environments in which it occurs’ [9]. Learning analytics comes in two forms [10]. The first is known as embedded analytics in which learners use real-time guided activities as part of their learning environment. The second is extracted analytics in which analysis of the learning environment is extracted and presented back to learners for interpretation separate to the original environment. Dashboards can be used as part of the learning analytics for both teachers and students, in an embedded or extracted form [11].

Dashboard applications extend to tracking data such as social interaction, resource use, activity and quiz results, works produced, and time spent on the system [11]. With such technological advancement, increasing amounts data can be tracked, leading to pedagogical interventions by teachers [12]. One such intervention is the use of dashboards for effective feedback to help with students’ self-regulated learning [13].

The literature in using dashboards for teaching in pharmacy is scarce. In pharmacy more generally, dashboards have been designed as part of the management of workloads and productivity [14] or as a part of cost-management [15]. Pharmacy education can benefit from applying learning analytics via dashboards, especially when building the communication skills of future clinicians. Thus, the aim of this project was to develop an interactive dashboard for the delivery of clinical skills feedback in the pharmacy curriculum.

## 2. Methods

This project followed an action-based research approach. Action based research involves putting ideas into practice as the means to developing ideas and improving practice [16]. Feedback from students on inconsistencies in marking was used as the initial stimulus to brainstorming the problem along with discussions with the tutoring team. After identifying the gap to be filled, the team collectively devised strategies, tested them in the classes, reflected on what worked, adjusted approaches, and added to or dropped features from the dashboard in an iterative fashion.

### 2.1. Setting

Three units of study at the University of Sydney were identified for the development of this tool. All units either involved patient counselling, or skills such as compounding. These units were identified as requiring a high level of student feedback provided from multiple tutors. This paper focuses on one course, dispensing and counselling in third year Bachelor of Pharmacy In 2020, 205 hundred students were enrolled in the dispensing and counselling course with 18 tutors, 5368 products were dispensed by students and 3385 counselling assessments were conducted. All products and assessments required the provision of timely, individualized feedback during the course.

### 2.2. Digital Ecology

The dashboard was part of a larger digital ecology used in the University of Sydney School of Pharmacy to facilitate teaching [17]. A set of digital artefacts were used to facilitate analysis, workspace, and communication amongst the teaching staff. In accordance with the TPACK model, the focus of this work is on technological and pedagogical knowledge to then integrate with the content knowledge of tutors derived from their practice experience, along with a tutor handbook to improve teaching consistency.

#### 2.2.1. Analysis

Two digital systems used were for analysis purposes. The first system used as a part of data capture was called the Student Relationship Engagement System (SRES^©^) [2]. SRES^©^ can be integrated with many learning management systems (LMS), for which we used Canvas LMS [18]. SRES^©^ was developed as a web-based system to house student data to be used to personalize interactions and support for students. For our subjects, we used it as a means of data capture for attendance, assessment and students’ activities. Tutors were able to enter the student task results and write feedback. These results were then transformed into our dashboard.

The visual dashboard was our second digital system for analysis. The data from SRES^©^ was downloaded via Python, and reports created using R and Rmarkdown. The dashboard was scheduled to update after each class, allowing enough time to have marks and feedback entered into SRES^©^.

#### 2.2.2. Workspace

The workspace was a storage unit for collaborative work and the task outcomes of the collaborations. Progressive notes to marking guides and feedback techniques would be updated in real-time. Google Docs^©^ was chosen as the platform for collaborative work as all tutors had access, the ability to limit access, and ease of use. Real-time edits by users would ensure information was always current. Google Sheets^©^ was also used to help with rostering tutors.

#### 2.2.3. Communication

Besides a virtual workspace for collaborating amongst teaching staff, an instant form of communication was also specified. Slack^©^ was used to allow conversation threads to be organized around each individual student activity. Conversations that were not pertinent to all staff could be discussed in different channels, reducing the notification count for all staff. With Slack^©^, staff no longer needed to provide a personal mobile number. The instant communication system facilitated consistency of marking, as well as providing tutor support in responding to individual student queries.

### 2.3. Evolution of the USyd Pharmacy Dashboard

An initial pre-dashboard design was implemented in 2017 as a static end-of-semester analytical report of student outcome for the individual classes. Providing a valuable snapshot of the year, a decision was made to create the USyd Pharmacy Dashboard to provide access to results throughout the semester, rather than at the end of the year. The measures to be included at the time were overall and class pass rates for each product.

The data was analyzed in R and the dashboard was made using Rmarkdown as a web page. In 2018, after each class, cohort analytics were generated, with sub-group analytics for individual classes. However, the single page output resulted in poor usability as a result of scrolling requirements.

An adaption incorporating visual dashboard design principles [19] was made in 2019 to present summary data, menus, and inter-class comparison metrics. The summary data display facilitated data integrity and quality assurance through visual presentation of percentage data entry completion.

A third USyd Pharmacy Dashboard design iteration was undertaken in 2020 to present inter-cohort comparison metrics between 2019 and 2020 student cohorts, thereby providing a real-time relative benchmark of cohort progress during the year. Table 1 shows the evolution of the USyd Pharmacy Dashboard and the features added over the course of development as the developers responded to requests and issues identified as the tool was used.

### 2.4. Evaluation

Student performance was measured by the proportion of students in each cohort that were able to demonstrate competency in the weekly clinical skills assessments. Student satisfaction was measured via the unit of study survey (USS) and frequency counts of requests for marking reviews. The USS is an annual student satisfaction survey for each unit of study measured on a 5-point Likert scale along with several free text entry questions allowing students to provide feedback on the unit of study in question. Counselling assessments are conducted one on one with a tutor, audio recorded, and rubric marks recorded in SRES. Students can request a review if they disagree with the outcome. A Chi-squared test was used to compare the USS scores, marking requests and collective assessments between 2019 and 2020.

## 3. Findings

### 3.1. USyd Pharmacy Dashboard in Action

For the unit of study coordinator, the USyd Pharmacy Dashboard has a number of practical applications.

1.Provides visual comparison of results between classes. This helps to identify classes where students may be having difficulty with the concepts being taught and allows resources to be directed to those classes, such as additional tutors.2.The visual comparison can identify discrepancies between how aspects of the course are being taught or assessed by different tutors. The communication tools can effectively distribute this information to tutors to improve consistency between tutors.3.Assess level of completion of data entry by the tutors and students.4.Provides feedback to students was part of the Canvas LMS integration with SRES. Student marks and tutor feedback was piped into the LMS system, allowing individual students to see the outcomes of the day’s activities. This was in real-time and reinforced learning if a student may have forgotten the feedback provided by the tutor.5.Students were not granted access to the live dashboard. However, at the end of each teaching week, the students were shown the summary group results. The weekly summary chart of results from both dispensing and counselling was used as the basis for feedback.

### 3.2. Current USyd Pharmacy Dashboard

The current USyd Pharmacy Dashboard starts with a ‘quick overview’ front page, as in Figure 1, providing the unit coordinator with an overview. The blue portion shows positive or preferred outcomes, with completion rates and passing students. The orange highlights the least desired aspects of the results. That is students not passing or scoring well. Gauges were used to observe activity completion rates. Completion rates could also be used to check data integrity.

Individual assessment items were observed afterwards. From Figure 2, the green represented the students count scoring perfect marks, orange was partial pass not causing harm to a patient and grey was unprofessional or unsafe products. Dips in scores from the overall cohort can help find difficult areas for students. The five lowest scores in Figure 2 can be attributed to changes in the task activities. Specifically, when an activity required more work than a process driven task, most students did not perform well. These tasks had either a clinical error that needed to be ratified, required communication with the doctor, or required extra details for the product labelling.

### 3.3. Dashboard Showing Time to Achieve Pass

A within cohort comparison allowed for comparisons of classes on the same day (Figure 3). As in Figure 4 there were 2 classes on the Thursday in 2019. From Figure 3, the first class outperformed the second class. All tutors stayed mostly consistent week-to-week. Therefore, it highlighted that these students in the second classes required more support. From the first few products in Figure 3, we can see that the second Thursday class was already struggling from the beginning and required additional teaching resources. A 2020 intervention was to provide tailored teaching resource allocation according to group performance.

A useful visual comparator was the time taken for students to achieve competency between cohorts. From Figure 4, we could see the number of students achieving competency over time. The vertical lines show the trajectory point at which students were deemed to have demonstrated competency in labelling and counselling assessments. The diagonal shows a perfect score for all students. We can see that students in 2020 were closer to the diagonal line, thus doing better than the students in 2019 (*X*^2^ = 108.22, *p* < 0.001). This signifies that the later 2020 cohort performed better than the 2019 in their dispensing and counselling activities. Visual comparison shows that the improved performance was consistent from the initial activities in the course.

The dashboard design and functions are available online at https://github.com/ardimirzaei/Pharmacy_Teaching_Dashboards, accessed on 19 April 2021. 

### 3.4. Evaluation

The unit of study survey (USS) contained 3 relevant questions. The first, “I have been guided by helpful feedback on my learning” answered on a 5-point Likert scale and then 2 open questions, “what have been the best aspects of this unit of study”, and “what aspects of this unit of study most need improvement?” Students have appreciated the improved consistency between tutors that has resulted from application of dashboard data. In the first year of teaching this unit, a common theme from the unit of study survey was dissatisfaction with discrepancies between tutors during assessments. These comments reduced in subsequent years. Overall, the USS average (out of 5) has shown steady improvement from the first year of implementation, 2018 (4.05, sd = 0.78) to the second year, 2019 (4.12, sd = 0.61) and most recently 2020 (4.38, sd = 0.62). From Figure 5, 2020 USS mean was statistically (*X*^2^ = 16.15, *p* < 0.0001) significant when compared to 2018 and 2019.

Requests for independent assessment reviews significantly declined over time; 16 requests in 2018, reducing to 8 in 2019 and zero in 2020 (*X*^2^ = 14.56, *p* < 0.001). During tutorial time, students have also appreciated seeing how they compare to other classes and being able to visualize the areas requiring improvement. They are then able to practice areas of weakness with their peers.

The unit of study coordinator reported significant time savings with regards to data entry, being able to identify areas of underperformance and streamlined communication with the teaching team.

## 4. Discussion

The USyd Pharmacy Dashboard was iteratively developed over a three-year period, allowing for the monitoring of student performance, staff consistency and maintaining data integrity. Student performance and satisfaction were demonstrated to have increased over this time. There is a paucity of literature on the use of dashboards in pharmacy education. A search of the Scopus database using the teams pharm* and dashboard did not provide any relevant studies. Literature was identified where dashboards were evaluated for their use in clinical decision making [20,21] and workflow management [22], however, these papers did not report on the use of dashboards to guide feedback to both students and tutors nor on the efficiency benefits provided. Thus, the USyd Pharmacy Dashboard is innovative in its application for the provision of feedback to both students and teachers. This approach is in accordance with the social pedagogy model described by McHugh et al. [23] whereby students and teachers develop both self-assessment and learner-centred coaching and encouraged seeking of constructive criticism in order to develop mastery.

Dashboards are an extremely useful tool from a teaching perspective, particularly in assessment (formative or summative) heavy units and teaching with multiple staff across several days. An element of the TPACK model combines knowledge of teaching pedagogy and curriculum content with technology [4]. Applying this element of the TPACK model through a student focused learning approach with adaptive feedback improves both learning and teaching. Technology is not an extra feature, but rather, a native element of our teaching and learning process. By monitoring student performance across days via the dashboard, the coordinator can recognize classes that may be underperforming compared to the rest of the cohort and investigate possible reasons. In our dispensing class we have identified instances of the tutor not emphasizing a counselling point of detail required in the dispensing process, for example the legislative requirements for cancelling a prescription for a drug of addiction. Using our communication channel, Slack, we can subsequently address this with the tutor concerned, prior to the next class. We have also identified classes that may have a higher-than-average number of students with no dispensing experience who are having more difficulties and have been able to direct more teaching resources to those classes. The USyd Pharmacy Dashboard assists in identifying incomplete marking or missing marks by showing a percentage complete for each class. When these are identified early, the tutors can be contacted and are more likely to have an accurate recollection of student’s marks.

A possible explanation of students’ performance could be the instructions provided to the cohort were clearer and resources were adaptively allocated to classes that appeared to be struggling. Furthermore, the tutors were marking more uniformly than previous years. Another possible explanation is that students in 2019 were sharing their notes to the 2020 cohort.

Lectures and tutorials can incorporate the visual representation of grading rubrics to show the class collectively areas that need further development. For example, a clear area of weakness in the communication grading was active listening and responding with empathy. By highlighting this and providing examples of how this can be done in practice, performance improved significantly in following assessments.

An action-based research approach allowed for an iterative development process. Regular feedback and discussion with the developer, coordinator and tutors was facilitated using Slack^©^ and allowed for prompt responses to issues raised.

### 4.1. Future Work

#### 4.1.1. Machine Learning Modelling for Exam Outcome

Machine learning models can be used to predict dropout and exam outcome [24,25,26]. Having a student’s progress in a subject, as well as their demographic and educational backgrounds can help build the required models to estimate student performance [27]. Educators will benefit from knowing students that are at risk of not completing a course, but also be able to detect high performers. These benefits include developing interventions early on to support low-performing students, as well as create further challenges for high performers [24]. Allocation of resources to students can be optimized through identify those with high and low needs.

#### 4.1.2. Real Time Benchmarking and Open Leaner Models (OLM)

The USyd Pharmacy Dashboard has the potential to be developed into an Open Learner Model (OLM). An OLM allows for data generated through the learning process to be made available to the learner, teacher and peers. The benefits of this model include encouragement of reflection, self-monitoring of performance, collaboration and competition with peers, thereby giving learners an increased level of control and responsibility for their learning [28]. OLMs can also be information hubs, incorporating links to resources. Finally, tasks can be adaptively made available to learners in response to either individual or collective performance.

#### 4.1.3. Integration of USyd Pharmacy Dashboard in SRES^©^/CANVAS

A further development is to have direct integration of the dashboard with an LMS for students to explore their progress and allow it to perform as part of an OLM. Students would be able to see their progress compared to others at the end of a workshop series. SRES^©^ can provide logic-based feedback and could provide set feedback and suggestions on how to improve based on the rubric responses.

#### 4.1.4. Time and Motion Study of Efficiency

It would be interesting to study the efficiency gains in terms of time spent on administrative tasks compared to similar units of study that do not use the USyd Pharmacy Dashboard as well as identify other time saving opportunities for unit of study coordinators. This would provide further evidence in favor of wider implementation of the dashboard.

#### 4.1.5. Use in Other Settings

Other educational activities such as large scale interprofessional learning activities at the University of Sydney, Faculty of Medicine and Health, may benefit from the introduction of the dashboard e.g., the introductory interprofessional education workshop [29] and the health collaboration challenge (HCC) [30]. Like the dispensing workshops, these involve large teams of tutors or demonstrators and large student cohorts. A visual dashboard to monitor student performance could be a valuable addition to these activities.

### 4.2. Strengths and Weaknesses

The strength in the way the USyd Pharmacy Dashboard was developed is the action-based research approach, where we could learn from our practical implementation and respond with changes in an ongoing fashion. As the coordinator and developer were both teaching and collaborating on the project, regular discussion and quick responses were possible. A weakness is that the perceptions of the tutors was not captured formally, however, future formal evaluation of the USyd Pharmacy Dashboard will take this into consideration.

## 5. Conclusions

Introduction of an iteratively developed visual dashboard facilitated applying the TPACK model for integrating technology, content, and pedagogy, in monitoring clinical skills education in an undergraduate pharmacy curriculum. This was associated with improved student performance and satisfaction. Students have appreciated the improved level of consistency and the feedback received from tutors in this unit of study. The unit of study coordinator reports increased level of efficiency and increased awareness of the data and can respond to improve teaching delivery and student experience. An action-based research approach has shown to be an effective method for approaching a practical problem-solving exercise with the benefits of flexibility and adaptability, allowing for customizable solutions that could be applied to other settings. Further refinement of the USyd Pharmacy Dashboard will continue throughout 2021 and formal evaluation is being planned.

## Figures and Tables

**Figure 1 pharmacy-09-00093-f001:**
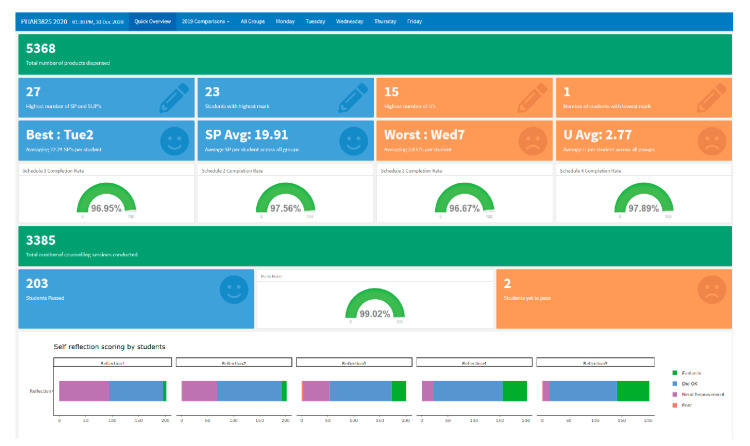
Current front page of the USyd Pharmacy Dashboard.

**Figure 2 pharmacy-09-00093-f002:**
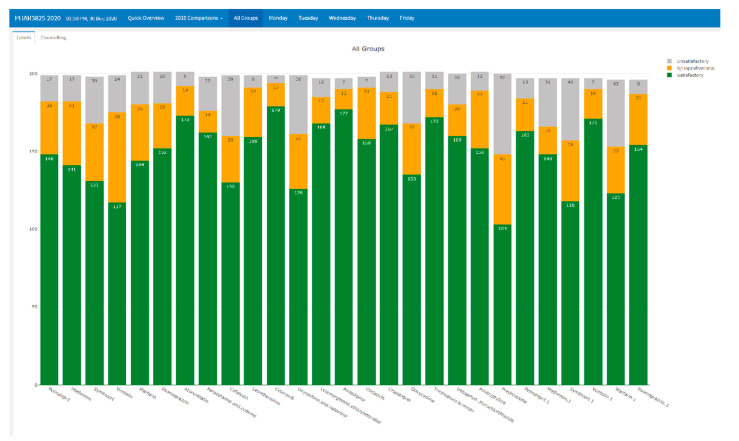
All group outcomes for 2020.

**Figure 3 pharmacy-09-00093-f003:**
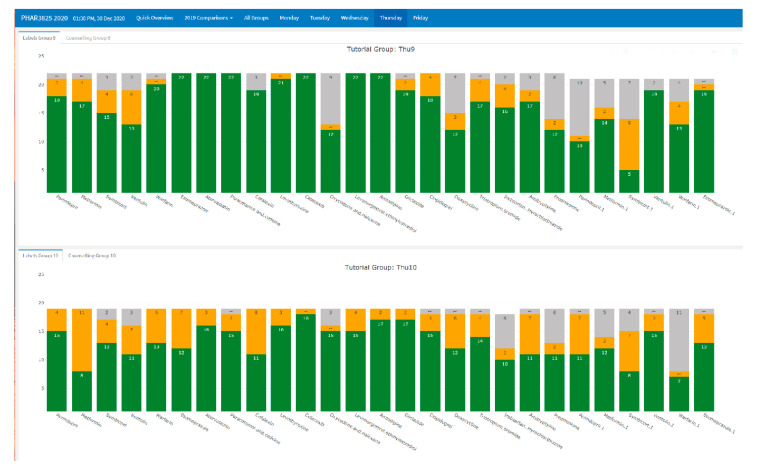
Comparisons between classes on the same day.

**Figure 4 pharmacy-09-00093-f004:**
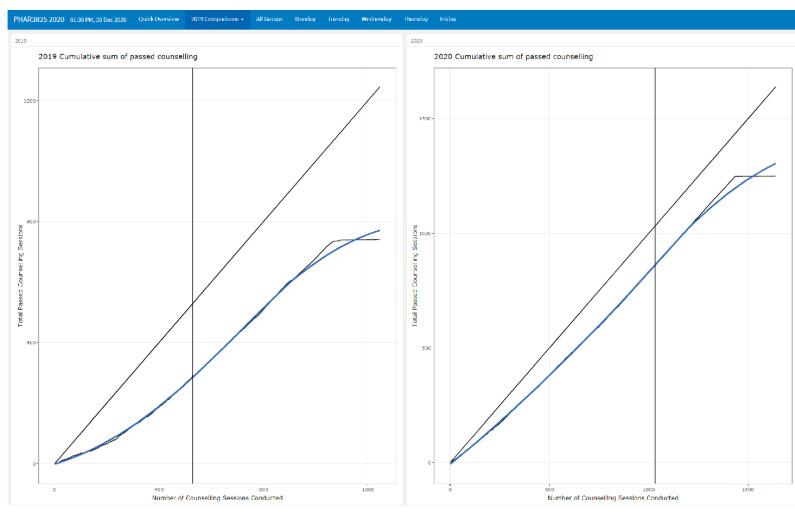
Comparisons tab comparing 2019 results with 2020.

**Figure 5 pharmacy-09-00093-f005:**
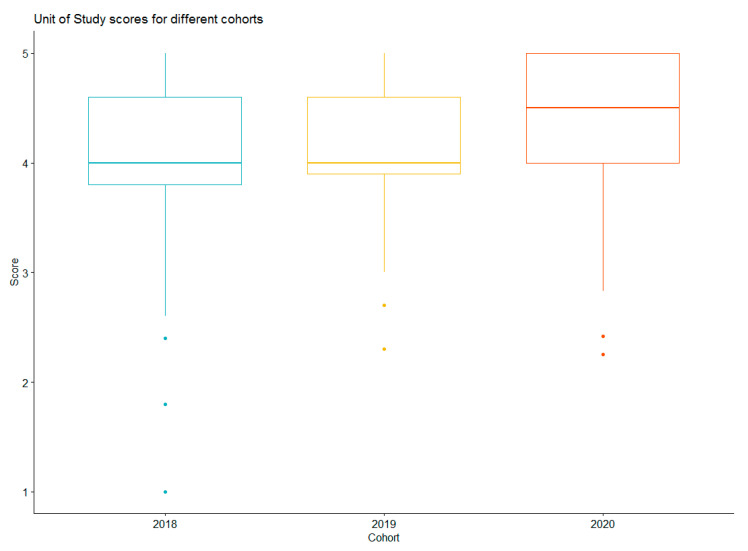
Comparisons of USS scores for the cohorts between 2018 and 2020.

**Table 1 pharmacy-09-00093-t001:** Evolution of the USyd Pharmacy Dashboard.

Measure	Implementation Year
Total pass rates per product	2018
Numerical tables for reasons for scoring low	2019
Comparator between classes	2019
Total products dispensed	2019
Total counselling conducted	2019
Data integrity gauge	2019
Self-reflection scores	2020
Comparator between cohorts	2020
Summary overviews	2020

## Data Availability

Please refer to suggested data availability statements in section “MDPI Research Data Policies” at https://www.mdpi.com/ethics.

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
