# Peer review of "Use of Visual Dashboards to Enhance Pharmacy Teaching"

_pharmacy, 2021, doi:10.3390/pharmacy9020093_

Round 1
Reviewer 1 Report
The manuscript describes the design and the process of development of a visual dashboard to enhance pharmacy teaching. The dashboard was iteratively developed over a three-year period following an action-based research approach. The development process would be even more convincing if a participatory design process (students, tutors and organizational staff) was used. The evaluation of the dashboard showed that student performance and satisfaction have increased over the last three years. Since the manuscript focused on describing and evaluating the dashboard, it does not contribute substantially to the field of research or to the existing literature about learning analytics.
Overall, I think the manuscript is interesting to the readers of pharmacy as it describes the concept of complex visual dashboard. Personally, I like the action-based research approach. The manuscript is well-written and the organization of the manuscript is clear. I miss a critical discussion of their work. It would be interesting to read about how the dashboard could be further improved (based on learning theories about feedback). What are the really innovative aspects (either on a technical or on a theoretical level).
Minor comments:
- the link https://github.com/ar-%20256dimirzaei/Pharmacy_Teaching_Dashboards does not work
- all figures are of poor quality
Reviewer 2 Report
Figures seem to be (at least in the evaluation version of the manuscript) a bit too low quality. It was a bit difficult to get data out of them.
I would like to see the evaluation questionnaire content of the student survey. Was the feedback collected with MCQ or open questions of both?
Conclusion is quite descriptive, I would like to see more in depth insight there.
Round 2
Reviewer 1 Report
I think the manuscript has been significantly improved and now warrants publication in Pharmacy. Congratulations!